# Blunted Cardiovascular Reactivity Predicts Worse Performance in Working Memory Tasks

**DOI:** 10.3390/brainsci13040649

**Published:** 2023-04-11

**Authors:** Brynja Björk Magnúsdóttir, Haukur Freyr Gylfason, Kamilla Rún Jóhannsdóttir

**Affiliations:** 1Department of Psychology, Reykjavík University, Menntavegur 1, 101 Reykjavik, Iceland; 2Department of Economics, University of Iceland, 101 Reykjavik, Iceland

**Keywords:** cardiovascular reactivity, cognition, psychological stress, blunted reactivity, behavioral disengagement, working memory

## Abstract

When we experience psychological challenges in the environment, our heart rate usually rises to make us more able to solve the task, but there is an individual difference in cardiovascular reactivity (CVR). Extreme CVR to environmental demands has been associated with worse health outcomes, with blunted CVR (little or no rise in heart rate) related to maladaptive behavior, including depression. The blunted CVR has been explained by motivational disengagement, which involves giving up on a task when facing obstacles. Disengagement is thought to be a habitual response that people might not be aware of, and, therefore, objective measures such as test performance might serve as a good measure of engagement. In this study, 66 participants solved different cognitive tasks while their CVR was measured. The aim was to test the association between test performance and reactivity, measured with the difference in heart rate at baseline and the mean heart rate while solving the tasks. Our results show a significant association between reactivity scores and performance in all tests, of various difficulty, indicating that blunted cardiovascular reactivity predicts poorer cognitive performance. Furthermore, we find an association between reactivity in one test and the performance in the other tests, suggesting that disengagement from environmental demands can be more general and not depend on the task at hand. The results, therefore, support earlier research suggesting that blunted CVR is associated with worse cognitive performance, and extends the literature by indicating that disengagement could be a more general maladaptive response to the environment.

## 1. Introduction

When people face psychological challenges, there is an individual difference in cardiovascular reactivity (CVR) that is associated with various diseases [1]. Extreme cardiovascular responses to acute psychological stress have been related to worse health outcomes, with heightened reactivity associated with, e.g., high blood pressure at rest [2], and cardiovascular risk status [3]. Therefore, the emphasis has been on dampening the cardiovascular response to protect cardiovascular health. More recently, the other end of the extremes, low cardiovascular reactivity to stress, has gained attention and has been associated with numerous disorders and maladaptive behavioral outcomes [4,5]. These include depression and anxiety [6,7], obesity [8], and substance abuse [9,10]. The association of extreme CVR and disease vulnerability is thought to reflect varied individual adaptability to environmental changes or challenges, suggesting that optimal cardiac regulatory capability predicts better health [11]. In line with that, the cardiovascular hypothesis states that better cardiovascular function can improve brain health [12]. For example, studies indicate that cognitive function improves after regular physical training, which has been attributed to greater blood flow to the brain [13]. Furthermore, brain activation increases while walking, which is thought to increase blood flow to the brain, even among people with neurological disorders and the elderly [14].

The main cardiovascular center is the medulla, which is a part of the brain stem, from where the sympathetic nervous system (SNS) can be activated to accelerate the heart rate and the parasympathetic nervous system (PNS), which slows the heart rate down [11]. The activation of these two systems can, therefore, be detected in our heart rate, and optimal cardiac regulation would be expected to activate these systems in response to the person’s environment, with a higher heart rate when solving cognitive tasks opposed to a lower heart rate during rest [11,15,16,17].

Blunted cardiovascular reactivity, a term used to describe a slight increase or even decrease in heart rate or blood pressure in response to stress, is thought to reflect dysregulation in fronto-limbic systems that control the autonomic system [4]. The fronto-limbic systems include the medial prefrontal cortex, anterior cingulate cortex, insula, hippocampus, and amygdala, and studies indicate changes in the function of these areas in response to stress [18], with reduced activation among people with blunted CVR [19]. It has been speculated that the dysregulation of the fronto-limbic system could be due to deficits in the physiological system, but more recently, a behavioral disengagement has been proposed as a likely explanation for the blunted CVR [4,20,21,22]. 

Disengagement involves giving up on a task when faced with obstacles and can also be seen as a lack of perseverance or coping skills [23,24]. Disengagement has been associated with depression and it has been suggested that task disengagement is a coping strategy learned in childhood as a defense mechanism to experienced threats [4,22,25]. Studies indicate that people with blunted CVR have met more adversity in childhood [26,27], though genetic factors also affect the individual difference in CVR to stress [28]. The defense mechanism might have been useful in childhood but results in negative health behavior in adulthood, such as depression [22]. Disengagement has been assessed with self-reports, and while Ginty et al. [29] found an association between blunted CVR and self-reported behavioral disengagement, other studies have not found an association between CVR and self-reported perseverance [30,31]. As engagement and disengagement are thought to reflect a function of the fronto-limbic system, a function that is habitual and people might not be aware of, it has been speculated that objective measures, such as cognitive test performance or perseverance on a task, might give better information on task engagement compared to self-reports [22]. 

Working memory has been defined as the ability to hold information temporally and integrate it with relevant information to guide behavior. Working memory is critical for high-level cognitive functions, such as problem-solving, reasoning, and goal-directed behavior [32,33]. Poor working memory has been associated with maladaptive behavior and various neurological and psychiatric disorders such as depression [34,35]. Furthermore, working memory capacity has been associated with learning style and how people use cues in the environment to guide behavior. It has been speculated that poor working memory could reflect more automatic responses that might result in maladaptive behavior, as opposed to the advantages of guiding behavior more effectively among those with more working memory capacity [36]. As the PNS should be activated in response to more environmental demands, we could expect higher heart rates when people use less automated responses. It would, therefore, be expected that heightened CVR, which produces greater blood flow to the brain, should result in better cognitive performance. Studies support that assertion, showing an association between CVR and cognitive function, with low reactivity associated with poorer cognitive performance [4,37,38], and less perseverance on an unsolvable puzzle task [31]. 

Within this framework, the blunted reactivity to stress could be seen as a marker of motivational dysregulation, rooted, at least partly, in childhood adversity, affecting coping strategies in adulthood and resulting in disengagement from challenges with poorer performance in cognitive tasks. Disengaging from a task has to do with giving up on task-related goals [4,22] and, as such, should directly affect the individual’s ability to perform working memory tasks that are highly goal-oriented. Prior work has, for the most part, examined how CVR to a highly stressful event relates to performance in another cognitive task. The aim of the present study is to expand on the current literature by testing the association between CVR during working memory tasks and performance in those same tasks. Therefore, the study further tests the hypothesis that blunted CVR when faced with a challenge is linked to task disengagement, resulting in poorer performance. Changes in heart rate from baseline to test conditions were measured using three different working memory tasks and measuring the task performance. It was hypothesized that changes in heart rate (increase or decrease compared with baseline) predicted performance in the working memory tasks, with blunted CVR predicting worse performance in all tasks.

## 2. Materials and Methods

### 2.1. Participants

An a priori power analysis was conducted to determine the required sample size for participants. A medium effect size of ƒ^2^ = 0.20 was chosen, with acceptable statistical power (1 − *β*) = 0.80, which indicated that a sample size of N = 65 was required [39,40]. A total of 66 staff members and students at Reykjavik University participated in the study, 41 females (62%) and 25 (38%) males. The participants’ age ranged from 19 to 55 (*M* = 28.2 years, *SD* = 8.8 years). Participants did not receive any credit for their participation. None of the participants reported any cardiovascular or neurological health problems. All reported normal or corrected-to-normal vision. This study received ethical clearance from the National Bioethics Committee in Iceland VSN-20-026-V2. Participants all read and signed an informed consent form before starting the experiment. 

### 2.2. Materials and Experimental Tasks

Participants were asked general questions regarding their age, gender, and whether they had any cardiovascular or neurological health problems. 

#### 2.2.1. Digit Span Test

Digit span is a measure of verbal working memory as well as of working memory capacity. The test is one of the subtests of the Wechsler Adult Intelligence Scale—Fourth Edition (WAIS-IV) [41]. Participants are asked to repeat a series of numbers in the first part and to repeat the numbers in the reverse order as well as in ascending order for the second part. Digit span has been reported to have good test–retest reliability (*r* = 0.83) and excellent internal consistency reliability (*α* = 0.93) [42]. In all sub-tests, the researcher began by repeating a sequence of three digits. If the subject correctly repeated each sequence of numbers, according to the instructions, with an equal number of digits in two consecutive occasions, then an additional number was added to the sequence, thus, increasing the difficulty of the task. However, when the participant failed to repeat the order of numbers correctly on two consecutive occasions, the researcher carried on to the next sub-task. The numbers on each sub-task were read to the participant at a rate of one word per second.

#### 2.2.2. Operation Span (OSPAN) Task 

The OSPAN task is a working memory task that measures the working memory span (capacity) by having participants solve simple equations and remember a word simultaneously [43,44]. Participants read an equation aloud (e.g., is (8 × 3) + 2 = 25) and answered whether the equation was correct or incorrect. The equation is followed by a word (e.g., car) that also is read out loud. The participants are presented with 12 sets of 3 × 2 equations/words, 3 × 3 equations/words, 3 × 4 equations/words, and 3 × 5 equations/words presented in random order. For each set (3 × 2, 3 × 3, 3 × 4, 3 × 5), there was an even distribution of one- to three-syllable words. The participant’s task is to remember the presented words in the correct order for each set. After each trial, participants were asked to write down the words from that trial in the order the words were presented. The score was based on the number of correct words remembered in the correct order. For example, if asked to remember “Phone, Statue”, a recollection of “Statue, Phone” would amount to 0 points. The total number of words to be remembered is 42; hence, the participants could obtain a maximum score of 42 and a minimum score of 0. Data from participants who solved less than 70% of the math problems correctly were excluded from further working memory analysis.

### 2.3. Apparatus

#### Finometer PRO

A beat-to-beat blood pressure monitoring system, Finometer PRO, was used for an online real-time measure of heart rate (HR) [45,46]. The Finometer PRO is a stand-alone, relatively non-invasive monitoring solution. The Finometer uses a finger cuff for measuring the pulse, but the absolute accuracy of the measured variables is calibrated using an upper arm cuff measurement. The system provides a continuous measure of HR at a sampling rate of 200 Hz. The data were exported and stored using a software program called Beatscope Easy. 

### 2.4. Procedure

The research was reported to the Data Collection Authority and authorization was obtained from the National Bioethics Committee. Data collection took place from May to September 2017 at Reykjavik University. Before starting the experiment, the participant was connected to the Finometer Pro. An upper arm cuff measurement was performed initially to calibrate the acquired data, but after that the participant was only connected to the system via a finger cuff that measured pulse continuously during the experiment. After calibrating the Finometer PRO, participants performed the digit span test (forward and backward), followed by the OSPAN test. 

### 2.5. Study Design and Data Analysis

The HR was collected continuously in real-time by the Beatscope Easy program, taking into consideration the participants’ height, weight, and age. Averages were calculated per minute for baseline, digit span (forward and backward together), and OSPAN. An average score was then calculated for each segment; three-minute baseline, digit span tasks, and OSPAN. The digit span test is relatively short compared to the OSPAN task, and we, therefore, used a single reactivity measure for both the digit span forward and backward tasks. Finally, difference scores were calculated by subtracting the baseline from digit Span and from OSPAN, providing two reactivity measures for each test. In order to test the contribution of reactivity as a continuous variable to performance in the OSPAN task and both the forward and backward digit span tests, hierarchical multiple regression models were conducted. Reactivity was entered on its own in step 1 and control variables (age, gender, BMI) were added in step 2. BMI was calculated based on the formula weight/height^2^ (*M* = 24.64, *SD* = 4.47). In addition, a correlation matrix was run to examine the bivariate correlation coefficient between the variables entered into the regression models.

## 3. Results

### 3.1. Descriptive Results and Correlations

Participants’ task performance is presented in Table 1. Performance scores on the OSPAN test range from 9–42. Performance scores on the digit span forward test range from 4–16 (*M* = 9.46, *SD* = 3.49), 2–12 on digit span backward (*M* = 6.68, *SD* = 2.63), and 9–42 on the OSPAN test (*M* = 23.29, *SD* = 9.66). Bivariate correlation coefficients were calculated for the variables entered into the regression models (Table 2). As can be seen in Table 2, both reactivity measures (for digit span and OSPAN) correlate with the relevant task. In addition, the reactivity measure for the digit span test correlates with performance in OSPAN and reactivity for the OSPAN test correlates with performance in digit span forward. The two reactivity measures are highly correlated. Age has a significant positive correlation with performance in the digit span tests, indicating that the older participants perform better on the tests. Gender is correlated with performance in the forward test, with women showing enhanced performance compared with men. Gender, age, and BMI have no significant correlation with OSPAN performance.

### 3.2. Hierarchical Linear Regression

The data were analyzed in hierarchical linear regression models, with the scores on the three memory tests as the dependent variables and the reactivity scores as the independent variables (step 1), and BMI, age and gender added as control variables in step 2 (Table 3). In the first step, reactivity predicts the performance in all the working memory tests, OSPAN (*F*(1, 61) = 7.80, *p* = 0.007), digit span forward (*F*(1, 54) = 6.03, *p* = 0.017), and digit span backward (*F*(1, 54) = 4.27, *p* = 0.044). Participants with higher reactivity scores show better performance in all the working memory tests. Reactivity explains a significant 11.3%, 10.0%, and 7.3% of the variance of the OSPAN, digit span forward, and the digit span backward tests, respectively.

Demographic characteristics (gender, age, and BMI) were added in the second step, without meaningful significant changes to the strength of the associations (*p* > 0.05). A case could be made that it is impractical to control for age and BMI, because those variables were used in the assessment of the reactivity score (i.e., controlling for them twice). However, since neither the magnitude nor the significance of the estimates change in a meaningful way, it is thought that it provides additional information on the strength of the associations between reactivity and the memory tests, given the strong correlation coefficient for age and digit span forward and digit span backward. The demographic characteristics explain a significant 37.1% and 39.3% of the variance of the digit span forward (Δ*F*(3, 51) = 11.93, *p* < 0.001) and the digit span backward tests (Δ*F*(3, 51) = 7.88, *p* < 0.001), respectively. However, the demographic characteristics do not add statisticall significance to the explanation of the variance of the OSPAN test (Δ*F*(3, 58) = 1.44, *p* = 0.24).

Figure 1 shows the results of the multiple linear regression, in which the results are depicted for average age female (27.2 years) with an average BMI of 23.9. The working memory test scores and the reactivity scores were standardized to aid interpretation. Across the three working memory tests, we find similar patterns of associations with reactivity. Reactivity correlates positively with all working memory test scores, controlling for gender, age, and BMI, such that participants with higher-than-average reactivity scores show better performance in the working memory tests.

## 4. Discussion

The aim of the present study is to further examine the hypothesis that blunted CVR is the result of task disengagement and is likely to lead to poor task performance and might indicate task disengagement. As predicted, reactivity while performing different working memory tasks correlates positively with scores on those tasks. The results are in line with earlier studies that have similarly found that blunted CVR is related to worse performance in various cognitive tasks [4,37,38,47,48]. Furthermore, reactivity measures for both tasks are highly correlated, and reactivity during one task correlates with the performance in the other task. Hierarchical linear regression shows that reactivity scores predict the outcome of the working memory tasks even after controlling for age, gender, and BMI (all variables affecting reactivity). 

Blunted CVR during task performance may reflect dysregulation in the fronto-limbic system [4,21], including diminished activity in the prefrontal cortex, which is thought to be highly involved in working memory, presumably resulting in worse performance in working memory tasks, as demonstrated in the present study. This is in line with previous work by both Chauntry [30] and Whittaker and Chauntry [31], which found that reactivity during a demanding working memory task (PASAT) predicted time spent on an insolvable puzzle task and time spent on a cold pressor task. Individuals with blunted response were quicker to give up on the tasks. 

The association between autonomic dysfunction as reflected in blunted CVR and cognitive performance is not well-understood or researched [29,48]. The most common set-up in the field measures reactivity during a highly stressful task, with the reactivity being the difference between baseline HR levels and HR during the stressful intervention. Reactivity is then linked to cognitive performance in different tasks. For example, Ginty et al. [48] tested HR reactivity during the performance in a highly demanding and stressful working memory task (PASAT), and looked at how well it correlated with a general intelligence test and performance in a choice reaction time test, measured 5 and 12 years later. They found that reactivity during PASAT correlated positively with cognitive performance at later time points measured on different tasks. However, following the claim that blunted CVR reflects task disengagement, it is important to examine the reactivity during a performed task and the potential association with performance in that particular task, as was performed in the present study. Task disengagement is related to giving up on task goals [4]. With highly goal-oriented tasks such as OSPAN, where the individual needs to keep track of multiple information and organize a response, task disengagement should be reflected in lower scores on the task. This is confirmed by the present results. Reactivity during the digit span task significantly predicts performance in both digit span forward and digit span backward. It also correlates significantly with the performance in the OSPAN task, supporting previous work showing that task disengagement as reflected in blunted CVR is associated with cognitive performance more generally.

Assuming that task disengagement is not an all-or-none response, it is interesting to look at the results of the present study where the degree of the extent of the blunted response is linearly related to task performance. The lower the difference between baseline and task values (less reactivity), the worse the performance, with those individuals showing negative values (less reactivity during task compared with baseline) with the worst performance in the three cognitive tasks. The results fit the biopsychosocial model in that task disengagement reflected in less, or no, reactivity is directly related to poor performance (e.g., Hase et al. [22]). Hase et al. (2020) [22] argue that blunted CVR cannot only be explained as the result of objective task difficulty. The present result supports this claim further, as the three tasks vary in their level of difficulty and none are extremely stressful, although they are challenging. Instead, the difference between individuals in terms of their CVR may be due to their coping mechanisms and how the task challenge at hand is evaluated. The results could also be seen within the framework of learning style [36] and how people with more working memory capacity can guide behavior more effectively while those with poorer working memory and blunted CVR could be disengaging from the task, using automated responses that demand less function in the fronto-limbic system, reflected in lower heart rate, and are less effective in meeting challenges in the environment. 

The results of the present study add to the broader claim that blunted reactivity may reflect a general behavioral disengagement [4] rather than being associated only with disengaging from acute stress. The disengagement may be linked to a maladaptive response to environmental demands, possibly due to poor coping strategies linked to some childhood trauma [4,22]. Furthermore, earlier research indicates blunted cardiovascular reactivity to environmental challenges among people with both depression and anxiety [7]. The association of the physiological response to environmental demands, disengagement, and mental health are not well-understood, but research indicating that an optimal cardiovascular response to challenges in the environment does produce both better cognitive performance and is predictive of better health [11], are of interest and need to be further researched.

A recent study by Benke et al. [20] indicated that blunted CVR could be an index of disengagement where less motivated (psychologically) individuals, during a FIFA soccer game, showed a more blunted HR response. However, blunted CVR was not associated with performance (goals scored vs. conceded goals), but after receiving feedback on poor performance, participants were likely to increase their performance in the next game and less likely to show blunted CVR. The authors suggest that healthy individuals might make use of poor performance and increase their motivational arousal to perform better next time [20]. It would be interesting to test if the same is true in a group of people with mental disorders and to study further reversing the motivational arousal among people with blunted HR reactivity as a means of treatment.

### Limitations and Future Studies

One of the limitations of this study was that we only used digit span forward and backward and OSPAN, but future studies should aim at using more varied cognitive tests measuring different cognitive factors. Furthermore, disengagement was not measured with self-reports, pupil size changes, or other direct measures. Instead, indications from earlier research suggested that blunted CVR and poor performance could be an indication of disengagement. It would be advisable for future research to add measures such as pupil size to be able to draw firm conclusions and to better understand the association of CVR, disengagement, and cognitive performance. 

In the current research, we only used a measure of heart rate, but it would have been useful to have further measures such as the galvanic skin response to indicate emotional arousal, to better understand people’s responses when solving cognitive tasks. Furthermore, we did measure heart rate at baseline and mean heart rate during individual tests, but we did not include resting periods between the three tests, which would have been an advantage for the study and interpretations of the study results.

Future studies could also look at more varied groups, as earlier research suggests the CVR is different between diverse groups. It would be of interest to further study people with maladaptive behavior such as substance use or people with depression. Furthermore, in this research we tested students and staff at a university, who are likely to show higher average performance in working memory tasks. Testing people with more varied working memory performance would be advisable. 

## 5. Conclusions

The results of this study indicate an association between task disengagement and cognitive performance, with blunted CVR predicting significantly poorer cognitive performance in all cognitive measures. While these results are in accordance with earlier studies, in the current study, we tested the reactivity with different tests and find cross-associations with performance in other cognitive measures. This could indicate an association between blunted CVR and cognitive performance in various settings, thereby lending support to the notion that blunted CVR could be the underlying mechanism of behavioral disengagement more generally. This disengagement could be seen as a maladaptive response to environmental demands rooted in childhood adversity where the response could have been seen as adaptive or a defense mechanism. In adulthood, when the disengagement response might have become habitual, the behavior could have different consequences and be maladaptive for the person, resulting in depression.

## Figures and Tables

**Figure 1 brainsci-13-00649-f001:**
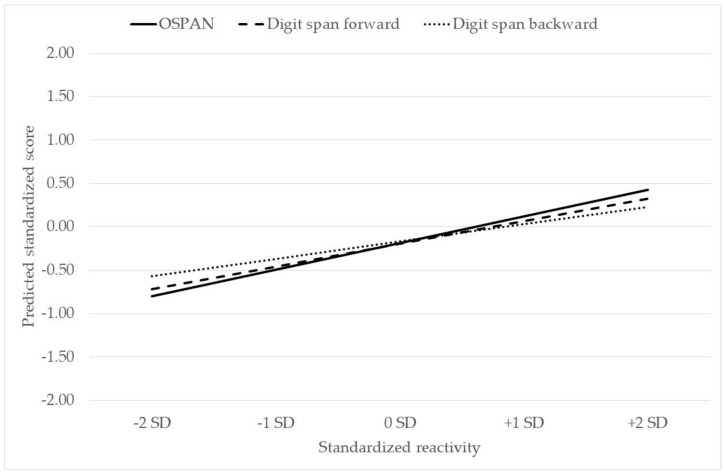
Predicted standardized scores for the three working memory tests (OSPAN test, digit span forward, and digit span backward) by standardized reactivity. High reactivity is indicated by two standard deviations (+2 *SD*) above the mean (0). Low reactivity is indicated to be two standard deviations (−2 *SD*) below the mean.

**Table 1 brainsci-13-00649-t001:** Means, standard deviations (*SD*), and minimum (Min) and maximum (Max) values for the task performance variables and reactivity.

	Mean	*SD*	Min	Max
DSF ^a^	9.46	3.50	4	16
DSB ^b^	6.68	2.63	2	12
OSPAN	23.29	8.66	9	42
React-digit span	5.51	5.49	−6.93	19.26
React OSPAN	3.74	5.06	−4.62	19.17

Note. ^a^ Digit span forward, ^b^ digit span backward.

**Table 2 brainsci-13-00649-t002:** Pearson’s bivariate correlation for task performance and potential predictor variables.

	1	2	3	4	5	6	7	8
1. DSF ^a^	1.0							
2. DSB ^b^	0.790 **	1.0						
3. OSPAN	0.344 **	0.426 **	1.0					
4. React-digit span	0.317 *	0.271 *	0.337 **	1.0				
5. React OSPAN	0.327 *	0.233	0.337 **	0.724 **	1.0			
6. Age	0.568 **	0.535 **	0.217	0.216	0.290 *	1.0		
7. Gender ^c^	0.274 *	0.233	0.230	−0.020	0.004	0.124	1.0	
8. BMI	0.220	0.139	0.005	−0.171	−0.101	−0.068	0.196	1.0

Note. ^a^ Digit span forward, ^b^ digit span backward, ^c^ female is the reference group. * *p* < 0.05, ** *p* < 0.01 (two-tailed).

**Table 3 brainsci-13-00649-t003:** Hierarchical linear regression results for the reactivity score predicting the working memory tests (OSPAN test, digit span forward, and digit span backward tests).

	OSPAN			Digit Span Forward			Digit Span Backward		
	B	SE b	*β*	B	SE b	*β*	b	SE b	*β*
Reactivity	0.576 **	0.206	0.337	0.201 **	0.082	0.317	0.129 *	0.062	0.271
Reactivity	0.525 **	0.214	0.307	0.166 **	0.068	0.263	0.095 *	0.056	0.200
Gender ^a^	3.793 *	2.144	0.216	1.034	0.767	0.141	0.642	0.632	0.117
Age	0.101	0.122	0.101	0.197 **	0.042	0.500	0.142 **	0.035	0.480
BMI	0.000	0.237	0.000	0.239 **	0.085	0.296	0.123 *	0.070	0.202
R^2^ (step 1)	0.113 **			0.100 **			0.073 *		
R^2^ (step 2)	0.175 **			0.471 **			0.367 **		

Note. ^a^ Female is the reference group. * *p* < 0.05, ** *p* < 0.01 (one-tailed in line with the reactivity hypotheses).

## Data Availability

Data are available on request.

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
