# Peer review of "Blunted Cardiovascular Reactivity Predicts Worse Performance in Working Memory Tasks"

_brainsci, 2023, doi:10.3390/brainsci13040649_

Round 1
Reviewer 1 Report
The study explores the relationship between CVR and performance in working memory tasks. Although well-written, there are significant issues need to be resolved.
- Data presentation is overly simplistic. For example, the authors should include raw heart rate data as a function of time to provide a more comprehensive understanding.
- The paper does not clearly explain the interval between the two tasks. It is essential to know why only one baseline heart rate was taken and if the heart rate returned to baseline after the first task. This information is crucial for interpreting the findings.
- While the Abstract, Introduction, and Discussion sections extensively discuss task disengagement, it is totally unclear how the task design reveals anything about task disengagement. Additional measures, such as self-report or pupil size, should be considered to provide more robust evidence of disengagement.
Author Response
We appreciate the opportunity to revise our manuscript, which we believe is improved after making the suggested edits. Below are your comments with our responses beneath each comment in bold letters, including how and where the text was modified when appropriate. We have made a major revision to the manuscript, and the changes are marked up using the track changes function.
_____________________
The study explores the relationship between CVR and performance in working memory tasks. Although well-written, there are significant issues need to be resolved.
1. Data presentation is overly simplistic. For example, the authors should include raw heart rate data as a function of time to provide a more comprehensive understanding.
We appreciate the comment and have now added extensively to the results chapter. We have added a table showing the means, standard deviations, min and max scores of raw scores for all the task performance variables and reactivity (see p. 5).
We did not plan for using the raw heart rate data and therefore were not able to use the raw scores as a function of time in this paper, however, we agree that future studies could focus more on pooled time-series analysis in order to provide a more complete picture of the association between CVR and the performance on working memory tasks.
2. The paper does not clearly explain the interval between the two tasks. It is essential to know why only one baseline heart rate was taken and if the heart rate returned to baseline after the first task. This information is crucial for interpreting the findings.
The standard practice is to take baseline at the beginning during total rest. Periods of recovery can be taken between different cognitive tests. This was not done in the present study and is often not done in the literature (for example, Gillie et al., 2014 and Thayer et al., 2019). However, we appreciate your point that the lack of a resting period between the two tasks is relevant for interpreting the findings. We have added to the discussion section that this could be a limitation to the study (see p. 8).
Gillie, B. L., Vasey, M. W., & Thayer, J. F. (2014). Heart rate variability predicts control over memory retrieval. Psychological Science, 25(2), 458-465.
Williams, P. G., Cribbet, M. R., Tinajero, R., Rau, H. K., Thayer, J. F., & Suchy, Y. (2019). The association between individual differences in executive functioning and resting high-frequency heart rate variability. Biological psychology, 148, 107772.
3. While the Abstract, Introduction, and Discussion sections extensively discuss task disengagement, it is totally unclear how the task design reveals anything about task disengagement. Additional measures, such as self-report or pupil size, should be considered to provide more robust evidence of disengagement.
We appreciate this comment, and we have added a number of citations with more thorough discussion on disengagement to the introduction, aiming to better explain how performance on a cognitive task, like working memory, could indicate engagement or disengagement. Furthermore, we discuss how an objective measure of behavior could give better information on engagement compared to self-assessment due to engagement being a habitual process that we might not be aware of (see p. 2).
We agree that measuring disengagement with pupil size would have been advisable, and we have added a discussion to the limitations section (see p. 8).
Reviewer 2 Report
The aim of the study was to investigate the hypothesis that blunted cerebrovascular reactivity (CVR) may indicate task disengagement and result in poor cognitive performance. The study found that reactivity while performing working memory tasks positively correlated with the scores on those tasks. The study also found that blunted CVR during task performance may reflect dysregulation in the fronto-limbic system and is associated with worse performance on working memory tasks. The study concluded that blunted reactivity may reflect a general behavioral disengagement and may be linked to a maladaptive response to environmental demands, possibly due to poor coping strategies linked to childhood trauma.
The data have been obtained from a well-designed study with a sufficient sample size. The statistical analysis is sound, the presentation of findings is well-structured and easy to follow. Limitations of the study are properly discussed. Overall, the manuscript represents a fine piece of scientific work.
The authors should consider the following comments during the revision to enhance the presentation of their findings:
1.What questionnaire was used to ask participants general questions?
2.What was the maximum and minimum score for the OSPAN task?
3.What variable was used in the present research from the Finometer PRO?
4.What were the results of the hierarchical linear regression models, and how did the reactivity scores predict performance on the WM tests after controlling for age, gender, and BMI?
5.What was the rationale for controlling for age and BMI in the hierarchical linear regression models?
Author Response
We appreciate the opportunity to revise our manuscript, which we believe is improved after making the suggested edits. Below are your comments with our responses beneath each comment in bold letters, including how and where the text was modified when appropriate. We have made a major revision to the manuscript, and the changes are marked up using the track changes function.
__________________
The aim of the study was to investigate the hypothesis that blunted cerebrovascular reactivity (CVR) may indicate task disengagement and result in poor cognitive performance. The study found that reactivity while performing working memory tasks positively correlated with the scores on those tasks. The study also found that blunted CVR during task performance may reflect dysregulation in the fronto-limbic system and is associated with worse performance on working memory tasks. The study concluded that blunted reactivity may reflect a general behavioral disengagement and may be linked to a maladaptive response to environmental demands, possibly due to poor coping strategies linked to childhood trauma.
The data have been obtained from a well-designed study with a sufficient sample size. The statistical analysis is sound, the presentation of findings is well-structured and easy to follow. Limitations of the study are properly discussed. Overall, the manuscript represents a fine piece of scientific work.
The authors should consider the following comments during the revision to enhance the presentation of their findings:
1. What questionnaire was used to ask participants general questions?
We did not use a published questionnaire to gather information on our participants but used standard questions about age, gender etc. This was not clear in the earlier version of our manuscript, but we have added information about the questions used to the methods section (see p. 3).
2. What was the maximum and minimum score for the OSPAN task?
We appreciate the comment and we have now added a table to the results chapter showing the means, min and max scores of raw scores for all the task performance variables and reactivity, including the OSPAN task (see p. 5)
3. What variable was used in the present research from the Finometer PRO?
The Finometer PRO provided us with a continuous measure of heart rate. This has now been made clearer in the sub-section 2.3.1 (see p. 4).
4. What were the results of the hierarchical linear regression models, and how did the reactivity scores predict performance on the WM tests after controlling for age, gender, and BMI?
The text in the results chapter was not clear enough in the previous manuscript. We have added extensively to the text in the new version of the manuscript (see pp. 5-6)
5. What was the rationale for controlling for age and BMI in the hierarchical linear regression models?
Prior studies have linked high BMI to blunted cardiovascular reactivity to a stressor (for example, Phillips, 2011). A meta-analysis by Chida and Hamer (2008) revealed that when looking at cardiovascular reactivity to stress (such as cognitive tasks), controlling for behavioral factors such as obesity had a significant impact on the results. Both cognitive performance and cardiovascular reactivity can also be sensitive to age. In the current study the participants' age range was 19 to 55 years.
Chida, Y., & Hamer, M. (2008). Chronic psychosocial factors and acute physiological responses to laboratory-induced stress in healthy populations: a quantitative review of 30 years of investigations. Psychological bulletin, 134(6), 829.
Phillips, A. C. (2011). Blunted cardiovascular reactivity relates to depression, obesity, and self-reported health. Biological psychology, 86(2), 106-113.
Reviewer 3 Report
The present research article by Magnusdottir and colleagues, entitled ‘Blunted cardiovascular reactivity predicts worse performance on working memory tasks’ is a well-written and useful summary on the status of knowledge of association between task disengagement and cognitive performance. Results showed that blunted cardiovascular reactivity (CVR) can predict significantly poorer cognitive performance on all cognitive measures.
The main strength of this manuscript is that it addresses an interesting and timely question, investigating the association between cardiovascular reactivity (CVR) during working memory tasks and performance on those same tasks. In general, I think the idea of this article is really interesting and the authors’ fascinating observations on this timely topic may be of interest to the readers of Brain Sciences. However, some comments, as well as some crucial evidence that should be included to support the author’s argumentation, needed to be addressed to improve the quality of the manuscript, its adequacy, and its readability prior to the publication in the present form, in particular reshaping parts of the Introduction and Methods sections by adding more evidence and theoretical constructs.
Please consider the following comments:
· A graphical abstract that will visually summarize the main findings of the manuscript is highly recommended.
· Keywords: Please consider adding ‘working memory’ as a keyword.
· Abstract: I believe that a lack of explanation of links between cardiovascular reactivity and stress, and how this physiological marker is altered in specific cognitive task engagement makes the reader unable to grasp the key aspects of this study by consulting directly the abstract.
· In general, I recommend authors to use more references to back their claims, especially in the Introduction of this meta-analysis, which I believe is lacking. Thus, I recommend the authors to attempt to expand the topic of their article, as the bibliography is too concise. Nevertheless, I believe that less than 50 articles are too low for a research article. Therefore, I suggest the authors to focus their efforts on researching relevant literature: in my opinion, adding more citations will help to provide better and more accurate background to this study.
· Introduction: The ‘Introduction’ section is well-written and nicely presented, with a good balance of descriptive text and information about the role of blunted reactivity to stress as a marker of motivation dysregulation. Neverthless, I believe that more information about neural substrates underlying cardiovascular reactivity, and how this controls and affect the autonomic system and therefore behavioral responses of individuals, will provide a better and more accurate background to this topic. Novel evidence have shown an extensive anatomical overlap between the distributed network of brain areas composing the central autonomic network (CAN), and the neural circuit critically involved in cognitive processes like emotional learning and working memory capacities (https://doi.org/10.1111/psyp.14122; https://doi.org/10.1038/s41598-019-43860-w). This information may help deepening information on how blunted cardiovascular reactivity may serve as an index of psychological task disengagement in the motivated performance.
· Materials and experimental tasks: I was wondering, did the authors take into the account the possibility to examine, together with changes in heart rate, skin conductance response?
· Results: In my opinion, this section is well organized, but it illustrates findings in an excessively broad way. Authors should provide better describe statistical information, rewriting this section more accurately, to ensure in-depth understanding of their findings.
· In my opinion, I think that a proper and defined ‘Conclusions’ paragraph would be very useful to state some thoughtful as well as in-depth considerations by the authors. In this section, Authors should make an effort, trying to explain the theoretical implication as well as the translational application of their research.
· In according to the previous comment, I would ask the authors to include a proper ‘Limitations and future directions’ section before the end of the manuscript, in which authors can describe in detail and report all the technical issues brought to the surface.
· Tables and Figures: According to the Journal’s guidelines, please provide a short explanatory caption for the table within the text. Also, I suggest to modify all figures for clarity because, as it stands, the readers may have difficulty comprehending it and to change the scale of the vertical axis and use the same minimum/maximum scale value in all the graphs.
I hope that, after these careful revisions, this paper can meet the Journal’s high standards.
I am available for a new round of revision of this paper. I declare no conflict of interest regarding this manuscript.
Best regards,
Reviewer
Author Response
We appreciate the opportunity to revise our manuscript, which we believe is improved after making the suggested edits. Below are your comments with our responses beneath each comment in bold letters, including how and where the text was modified when appropriate. We have made a major revision to the manuscript, and the changes are marked up using the track changes function.
__________________
The present research article by Magnusdottir and colleagues, entitled ‘Blunted cardiovascular reactivity predicts worse performance on working memory tasks’ is a well-written and useful summary on the status of knowledge of association between task disengagement and cognitive performance. Results showed that blunted cardiovascular reactivity (CVR) can predict significantly poorer cognitive performance on all cognitive measures.
The main strength of this manuscript is that it addresses an interesting and timely question, investigating the association between cardiovascular reactivity (CVR) during working memory tasks and performance on those same tasks. In general, I think the idea of this article is really interesting and the authors’ fascinating observations on this timely topic may be of interest to the readers of Brain Sciences. However, some comments, as well as some crucial evidence that should be included to support the author’s argumentation, needed to be addressed to improve the quality of the manuscript, its adequacy, and its readability prior to the publication in the present form, in particular reshaping parts of the Introduction and Methods sections by adding more evidence and theoretical constructs.
Please consider the following comments:
- A graphical abstractthat will visually summarize the main findings of the manuscript is highly recommended.
We appreciate the suggestion but are not able to produce graphical abstract at this point.
- Keywords: Please consider adding ‘working memory’ as a keyword.
Thanks for the suggestion; we have added working memory as a keyword (see p.1).
- Abstract: I believe that a lack of explanation of links between cardiovascular reactivity and stress, and how this physiological marker is altered in specific cognitive task engagement makes the reader unable to grasp the key aspects of this study by consulting directly the abstract.
We do agree that an explanation was lacking in the previous version of the manuscript. We have now added to the text in our abstract with the aim of helping the reader to grasp the main aspects of the study from the abstract (see p.1).
- In general, I recommend authors to use more references to back their claims, especially in the Introduction of this meta-analysis, which I believe is lacking. Thus, I recommend the authors to attempt to expand the topic of their article, as the bibliography is too concise. Nevertheless, I believe that less than 50 articles are too low for a research article. Therefore, I suggest the authors to focus their efforts on researching relevant literature: in my opinion, adding more citations will help to provide better and more accurate background to this study.
Thanks for the suggestion. We have added significantly to the text, especially in the introduction chapter, with more citations to provide a better background for our study (see p. 2).
- Introduction: The ‘Introduction’ section is well-written and nicely presented, with a good balance of descriptive text and information about the role of blunted reactivity to stress as a marker of motivation dysregulation. Neverthless, I believe that more information about neural substrates underlying cardiovascular reactivity, and how this controls and affect the autonomic system and therefore behavioral responses of individuals, will provide a better and more accurate background to this topic. Novel evidence have shown an extensive anatomical overlap between the distributed network of brain areas composing the central autonomic network (CAN), and the neural circuit critically involved in cognitive processes like emotional learning and working memory capacities (https://doi.org/10.1111/psyp.14122; https://doi.org/10.1038/s41598-019-43860-w). This information may help deepening information on how blunted cardiovascular reactivity may serve as an index of psychological task disengagement in the motivated performance.
We appreciated the suggestion, and we believe the Garofalo et al. study helped explain our topic better. In addition, we added further discussion that we thought might explain this phenomenon better (see p. 2 and pp. 7-8).
- Materials and experimental tasks: I was wondering, did the authors take into the account the possibility to examine, together with changes in heart rate, skin conductance response?
Thanks for the suggestion. It would definably have been useful to measure the galvanic skin response in addition to HR and we have added this suggestion to Limitations (see p. 8).
- Results: In my opinion, this section is well organized, but it illustrates findings in an excessively broad way. Authors should provide better describe statistical information, rewriting this section more accurately, to ensure in-depth understanding of their findings.
The text in the results chapter was not clear enough in the previous manuscript. We have added extensively to the text in the new version of the manuscript and we have added a table showing the means, standard deviations, min and max scores of raw scores for all the task performance variables and reactivity (see pp. 5-6).
- In my opinion, I think that a proper and defined ‘Conclusions’ paragraph would be very useful to state some thoughtful as well as in-depth considerations by the authors. In this section, Authors should make an effort, trying to explain the theoretical implication as well as the translational application of their research.
We have rewritten the conclusion section with the aim of explaining our results better to the reader (see pp. 8-9).
- In according to the previous comment, I would ask the authors to include a proper ‘Limitations and future directions’ section before the end of the manuscript, in which authors can describe in detail and report all the technical issues brought to the surface.
Thanks for the suggestion. We have added a proper Limitations and future directions section into the revised manuscript and added considerably to the text (p. 8).
- Tables andFigures: According to the Journal’s guidelines, please provide a short explanatory caption for the table within the text. Also, I suggest to modify all figures for clarity because, as it stands, the readers may have difficulty comprehending it and to change the scale of the vertical axis and use the same minimum/maximum scale value in all the graphs.
We appreciate this suggestion and have added “Note” to all tables (1, 2 and 3). We have also changed the scale of the vertical axis of figure 1 showing the same minimum/maximum values for both axes. And we changed the font in the picture to be consistent with the font in the manuscript (see pp. 5-6)
Round 2
Reviewer 1 Report
Thank the authors for addressing my concerns. The authors added more data and discussed further limitations of the study, which has made the paper stronger. I have no more comments.
Reviewer 3 Report
The authors did an excellent job clarifying all the questions I have raised in my previous round of review. Currently, this paper entitled ‘Blunted cardiovascular reactivity predicts worse performance on working memory tasks’, is a well-written, timely piece of research that described the association between cardiovascular reactivity (CVR) during working memory tasks and performance on those same tasks.
Overall, this is a timely and needed work. It is well researched and nicely written, therefore I believe that this paper does not need a further revision.
I am always available for other reviews of such interesting and important articles.
Thank You for your work, Reviewer